# Evaluating the Genome-Based Average Nucleotide Identity Calculation for Identification of Twelve Yeast Species

**DOI:** 10.3390/jof10090646

**Published:** 2024-09-11

**Authors:** Claudia Cortimiglia, Javier Alonso-Del-Real, Mireya Viviana Belloso Daza, Amparo Querol, Giovanni Iacono, Pier Sandro Cocconcelli

**Affiliations:** 1Department for Sustainable Food Process (DISTAS), Università Cattolica del Sacro Cuore, 29122 Piacenza, Italy; claudia.cortimiglia@unicatt.it (C.C.); mireyaviviana.bellosodaza@unicatt.it (M.V.B.D.); 2Tuberculosis Genomics Unit, Instituto de Biomedicina de Valencia (IBV-CSIC), 46010 Valencia, Spain; jalonso@ibv.csic.es; 3Grupo de Biología de Sistemas en Levaduras de Interés Biotecnológico, Departamento de Biotecnología de los Alimentos, Instituto de Agroquímica y Tecnología de Los Alimentos (IATA-CSIC), 46980 Valencia, Spain; aquerol@iata.csic.es; 4European Food Safety Authority (EFSA), 43126 Parma, Italy; giovanni.iacono@efsa.europa.eu

**Keywords:** yeast species delineation, FastANI, whole genome sequencing, multi-loci characterization, taxogenomics

## Abstract

Classifying a yeast strain into a recognized species is not always straightforward. Currently, the taxonomic delineation of yeast strains involves multiple approaches covering phenotypic characteristics and molecular methodologies, including genome-based analysis. The aim of this study was to evaluate the suitability of the Average Nucleotide Identity (ANI) calculation through FastANI, a tool created for bacterial species identification, for the assignment of strains to some yeast species. FastANI, the alignment of in silico-extracted D1/D2 sequences of LSU rRNA, and multiple alignments of orthologous genes (MAOG) were employed to analyze 644 assemblies from 12 yeast genera, encompassing various species, and on a dataset of hybrid *Saccharomyces* species. Overall, the analysis showed high consistency between results obtained with FastANI and MAOG, although, FastANI proved to be more discriminating than the other two methods applied to genomic sequences. In particular, FastANI was effective in distinguishing between strains belonging to different species, defining clear boundaries between them (cutoff: 94–96%). Our results show that FastANI is a reliable method for attributing a known yeast species to a particular strain. Moreover, although hybridization events make species discrimination more complex, it was revealed to be useful in the identification of these cases. We suggest its inclusion as a key component in a comprehensive approach to species delineation. Using this approach with a larger number of yeasts would validate it as a rapid technique to identify yeasts based on whole genome sequences.

## 1. Introduction

The species delineation of yeasts, intended as the process of assigning a yeast strain to a specific taxonomic group and of establishing the boundaries between different yeast species, is still a complex endeavor. Since the beginning of microbial taxonomy, for both prokaryotic and eukaryotic microorganisms, different meanings have been attributed to the ‘species’ concept and this evolution has been linked to the increase in knowledge about biological and genomic aspects of microbes. Alongside the evolution of the species concept, various methods have been developed to effectively define what is meant by a species. Although DNA-based molecular analysis improved the distinction between yeast species, by using several markers [1], such as the variable domains D1 and D2 of the nuclear large subunit ribosomal RNA (LSU rRNA) gene and internal transcribed spacer (ITS) sequences [2], in the case of some closely related species, their simultaneous use was demonstrated not to be sufficiently discriminating [3]. For this reason, these universal markers have been associated with other genetic markers, typical of each genus and species, constituting the so-called polyphasic approach [4]. Some examples of other markers are cytochrome oxidase II (*COX II*) [5], translation elongation factor 1α (*EF-1α*) (Kutzman and Robnett, 2003), RNA polymerase II gene [6], and the actin gene [7]. The alignment of concatenated genes is used to infer phylogenetic relationships and deduce the belonging to a determined species. Molecular analysis of a combined dataset of marker genes is currently integrated with the polyphasic approach for the determination of strain conspecificity, including not only molecular investigation but also a phenotypic evaluation. Indeed, nowadays, the species concept is built by looking at several aspects of a strain, starting from the so-called Phenotypic Species Concept (PhenSC), which includes phenotypic properties, like morphological characteristics of colonies and growth requirements in terms of substrates and temperature, and taking in consideration the Phylogenetic Species Concept (PSC), determined on the genetic distance calculated using different barcodes [8].

With the advent of genomic technologies, yeast taxonomy and characterization have undergone a transformative shift [9]. Genomic sequences have emerged as invaluable resources, providing a wealth of information for unraveling the intricate relationships among yeast taxa and species. In particular, whole-genome sequencing (WGS) and analysis allowed for a more comprehensive and accurate understanding of many of their biological features by studying intra-genus [10,11] and intraspecies variations [12,13,14,15,16], including the delimitation of species complexes [17], the complexity of population structure [18], evolutionary scenarios [19,20], domestication events [21], and geographic adaptation [18].

The latest discoveries in yeast genetics and genomics have also helped to deepen our knowledge of the evolutionary capabilities revealed through hybridization, particularly within the *Saccharomyces* genus [22,23]. The occurrence of past hybrid matings, accompanied by subsequent backcrossing and other postzygotic processes, has resulted in an imbalance in the relative contributions of parental genomes. This event makes species delineation and identification more complicated. For example, although LSU rRNA is used extensively for species identification, it is known that intragenomic variations in this region can arise, presenting challenges in achieving precise and accurate species identification [24,25]. 

In bacteria, the application of WGS has led to the development of a number of metrics to assess species delineation. The systematic process to identify species is performed by the overall genome relatedness index (OGRI), which estimates the relatedness between the genome of a strain to be identified and the genome of a type strain. Among the algorithms for OGRI calculation, the most widely used is Average Nucleotide Identity (ANI), which measures the genetic identity through the pairwise comparison of coding regions within genomes, and digital DNA-DNA hybridization (dDDH), namely the calculation of overall in silico similarity between the genomes of two strains, integrated with the 16S rRNA sequence similarity analysis [26]. For bacteria, boundaries of ≈94–96% and 70% for ANI and dDDH, respectively, are commonly accepted for species delimitation. The application of this metric to yeast is not particularly widespread and still needs to be explored in depth. Lachance and colleagues [27] were the first to use an ANI approach to delineate *Metschnikowia* species, comparing the results they obtained with this approach with a phylogenetic tree built on the 100 largest orthologs and suggesting 95 ± 0.5% as the threshold for species circumscription [28]. 

The aim of this work was to define ANI values to be used to set species boundaries for the taxonomic identification of yeast strains and for the delineation of some yeast species. For this purpose, we used FastANI, a bioinformatics tool that calculates ANI of orthologous genes using alignment-free workflow with a significantly improved analysis speed, originally designed for bacterial species identification [29]. To assess the performance of this method, we tested a dataset of 644 assemblies of yeast genera of biotechnological interest; the results were then compared to two other methods, one based on the alignment of D1/D2 region of LSU rRNA extracted in silico and the other based on the multiple alignment of orthologous genes (MAOG) among the species in question. Moreover, to assess the use of FastANI in the hybrid strains, we compared the three methodologies using 77 previously published hybrid strain sequences of *Saccharomyces* [30,31,32,33] and 150 genomes artificially simulating introgression created for the present work.

## 2. Materials and Methods

### 2.1. Yeast Assemblies Used in This Study

Twelve yeast genera belonging to *Saccharomyces*, *Debaryomyces*, *Rhodotorula*, *Hanseniaspora*, *Kazachstania*, *Kluyveromyces*, *Pichia*, *Starmerella*, *Torulaspora*, *Yarrowia*, *Candida*, and *Schizosaccharomyces* were selected. All except *Candida* and *Schizosaccharomyces* were chosen because of their numerous applications in biotechnology. We considered *Candida* as a genus of clinical importance as well as having biotechnological potential [34]; we included *Schizosaccharomyces* as an outlier in the taxonomic rank because it belongs to a different Order. Within each genus, all available assemblies from all species were downloaded from the NCBI Genome repository except for known hybrid species.

Since the number of available draft genomes for *Saccharomyces cerevisiae* was more than 1000, we decided to consider a restricted number of them; we selected the 79 draft genomes analyzed by Legras and coworkers, which reflect the genetic variability of the species [35]. 

Overall, a total of 644 assemblies deposited in the NCBI Genome database and classified as the above-mentioned genera were downloaded to be part of the dataset. These assemblies showed different completeness statuses, as reported in Figure 1, and quality. The scaffold assembly level was the most represented (64.3%), while complete and chromosome status were less common.

Appendix A provides a list of the considered genomes, with details on the genus, species, accession number, indication of reference strains, identification of CBS strains, sequencing technology, coverage, and BUSCO-based completeness.

We created a dataset with 77 *Saccharomyces* interspecific hybrid strains selected from previous publications [30,31,32,33] (Appendix A), downloading SRA files from the NCBI Sequence Read Archive. Reads were trimmed for adapter removal and quality filtered with Trimmomatic [36] and de novo assembled with SPAdes [37]. Furthermore, the proportion of reads that mapped against the different parental species of each hybrid was calculated with sppIDer [38].

### 2.2. Species Delineation Using ANI Calculation, In Silico Alignment of D1/D2 Region of LSU Ribosomal DNA, and the Selection of Orthologous Genes (MAOG)

Three methods were used to obtain identity values between assemblies within each genus. One is based on the alignment-free ANI calculation; the second is based on the in silico extraction of ribosomal barcode sequence D1/D2 from assemblies; and the third employs a multiple sequence alignment (MAOG). The scheme of the applied methodology is described in Figure 2.

The assemblies downloaded from the NCBI Genome database were analyzed with FastANI v1.32 [29]. FastANI is specifically designed to estimate ANI within the 70–100% identity range. This configuration enables FastANI to generate mappings with alignment identities that are close to 70% or higher. The analysis was run using the following parameters: fragment length of 3000, kmer size of 16, and minimum fraction of shorter genome coverage of 50%. It was applied to assemblies belonging to different species within each genus separately, with an “all against all” approach, using the same genomes as queries and references and using the—matrix parameter. The outputs consisted of matrices containing the identity percentage between genomes of strains in the same species. 

Barrnap v0.9 (https://github.com/tseemann/barrnap) (accessed on 7 September 2023) was used to predict and extract 28S sequences from the considered assemblies. From these, D1/D2 regions of LSU ribosomal DNA were extracted using in silico PCR (https://github.com/simonrharris/in_silico_pcr) (accessed on 20 September 2023). Primers were NL1-forward (5′-GCATATCAATAAGCGGAGGAAAAG-3′) and NL4-reverse (5′-GGTCCGTGTTTCAAGACGG-3′) or LR6-reverse (5′-CGCCAGTTCTGCTTACC-3′). The extracted D1/D2 sequences were then classified according to the genus they belonged to according to the NCBI Genome database and aligned with MAFFT v7.390; identity values were calculated from these alignments.

For the MAOG method, assemblies were analyzed with BUSCO v5.2.2 [39]. The Fungi Odb10 database was used to extract orthologous that are conserved in the kingdom Fungi, with the AUGUSTUS gene predictor parameter on. Assemblies resulting in fewer than 200 orthologous genes annotated as “single complete copy” by BUSCO were filtered out. The remaining assemblies were classified according to their genus in the NCBI Genome database. The intersection of orthologous sequences identified in all the assemblies of each genus was then listed. Within each species, the sequences of all these filtered orthologs were put together into a multi-FASTA file for every ortholog. For each of these files, nucleotide sequences were translated and aligned with MAFFT multiple alignment software v7.390 [40]. The alignments were then back-translated and trimmed with TrimAl v1.4.1 (http://trimal.cgenomics.org/) (accessed on 12 September 2023). Subsequently, all orthologous gene alignments of each genus were concatenated into a single file, with as many records as there were available assemblies for that genus. Each record contained the aligned sequence of each of the orthologs in the list for that genus, one after the other. The square root of the pairwise distances (D) between these alignments was calculated using the dist.alignment function (parameter matrix set to identity) of the adegenet R package v1.7. Then, pairwise distances were transformed into identity values by calculating the difference in 1 minus the squared value of D.

### 2.3. Statistical Analysis for the Comparison between D1/D2 Sequence, MAOG, and ANI Calculation

Outliers were assumed to be wrongly identified species and were corrected. Thus, assemblies with a FastANI identity value above 99%, D1/D2 identity value above 99%, and alignment-based identity value above 95% were considered to belong to the same species, and assemblies with a FastANI identity value below 90% and an alignment-based identity value below 90% were designated as different species. Boxplots were obtained with the ggplot2 R package v2 3.4.2 function geom_boxplot, taking into account the Boolean variable of whether two assemblies were from the same species or not. Density plots of the identity values obtained with each of the three methods were obtained with the R base function density. In addition, we obtained the local minima for each method, both for the full dataset and the *S. cerevisiae* dataset, from density distributions of identity values computed by the function density from the R base. A paired *t*-test was carried out by applying the Welch correction due to differences in variance distribution between groups of samples on the identity values from either MAOG or D1/D2 sequence analysis versus the identity values from FastANI for the same species assembly comparisons. Moreover, a Cohen’s D test was performed to evaluate the effect size. PCA was performed with the R function prcomp v4.2.1.

## 3. Results

### 3.1. Species Delineation Based on ANI Calculation

The first objective was to assess how effective FastANI analysis was in identifying strains belonging to the same species and discriminating genomes belonging to different species, considering those reported in NCBI to be valid. 

When FastANI was run on the dataset composed of 644 assemblies, the tool provided pairwise comparison values for all genome pairs tested (complete data are shown in Appendix A). Regardless of genus and species, the similarity identities obtained through FastANI proved effective in distinguishing between different species. Different cutoffs were defined depending on genus, but overall, if two genomes showed identity values below 90%, they belonged to separate species, as described in detail below. Differently, genome pairs showing values above 94–95% were classified among the same species. Table 1 gives a detailed breakdown of the cutoffs defined for each species analyzed within each individual genus.

*Candida*. Within the genus *Candida*, we analyzed a total of 114 genomes, including 12 species. Where genomes belonged to the same species, the similarity identity exceeded 96%, while ANI values between strains of different species were below 88%. The most represented species, *C. albicans* (comprising 70 assemblies), showed intraspecies ANI values above 98%. Since the identity exceeds 96% between strains of the same species within the *Candida* genus, this value is proposed as the cutoff for the species delineation. An ANI value of 92% between *C. sanyaensis* and *C. sojae* was observed. This intermediate value, falling between the threshold for the delineation of the same or different species, suggests a greater similarity between these two species that may be the result of a hybridization event.

*Debaryomyces*. Within the *Debaryomyces* genus, 19 assemblies encompassing seven species were analyzed (Figure 3A, Appendix A). FastANI demonstrated effective discrimination between species, with pairwise ANI values consistently falling below 90%. Discrepancies were identified in certain assemblies belonging to the *D. hansenii* species: assemblies GCA_006942225.1 and GCA003349505.1 displayed low identity values (below 90%) with other genomes belonging to *D. hansenii* species, including the reference genome of CBS767T, but exhibited high ANI values (99.41% and 99.45%, respectively) with *D. fabryi*. Similarly, strains MTCC 234 (GCA_000239015.2) and J6 (GCA_001682995.1) originally identified as *D. hansenii* showed very high ANI values (99.92% and 99.95%, respectively) with *D. subglobosus* and between each other (100% and 99.92%, respectively). These comparisons suggest a potential misidentification of the isolates. 

The *D. hansenii* assemblies GCA_006942205.1, GCA_006942215.1, GCA_006942225.1, GCA_006942235.1, and GCA_006942305.1 showed reciprocal ANI values exceeding 95%, indicating that they belong to the same species. However, they displayed lower ANI values (approximately 92%) than the other three assemblies of the same species (GCA_006942325.1, GCA_016097515.1, and GCA_016097625.1). A pairwise identity of 92% may be characteristic of the *D. hansenii* species or could be attributed to complex genomic alterations such as hybridization with closely related species. Higher identity percentages with other stains among the same genus *Debaryomyces* were not observed.

*Hanseniaspora*. Within *Hanseniaspora* genus, 17 species were deposited in the NCBI Genome database, with a total number of 31 genomes. *H. uvarum* was the species with the highest number of deposited assemblies. The application of FastANI to the *Hanseniaspora* genus demonstrated the tool’s effectiveness in delineating species boundaries, evident in ANI values below 90% between different species (Figure 3B, Appendix A). ANI values exceeding 97% were consistently observed among genomes belonging to the same species. However, a noteworthy exception involves two genomes of *H. occidentalis* exhibiting a reciprocal ANI value of 91.84%. This inconsistent result was due to a misidentification during the sequencing of the D1/D2 region [17,41].

*Kazachstania*. This genus consisted of 28 assemblies, grouped in 21 species. Although few genomes belonging to the same species were available for this genus, FastANI analysis evidenced highly discriminatory values among the different species of *Kazachstania* genus. ANI values below 89% effectively discriminated between various species, with the exception of *K. exigua* and *K. turicensis*, which displayed a remarkably high identity percentage of 99.86%. This suggests that these two species likely belong to the same taxonomic unit. Since more than one genome was available only for *K. unispora*, *K. barnettii,* and *K. servazzii*, the proposed cutoff that determines the same species can be applied only to them. In particular, ANI cutoff values exceeding 98% can be defined for the delineation of the species.

*Kluyveromyces*. The 27 assemblies downloaded from the NCBI Genome dataset belonged to seven species. Of these, three were represented by a single assembly, while for the remaining four species, two or more assemblies were analyzed. The intraspecies ANI value exceeding 95% was observed among the *K. marxianus* assemblies, which were the most abundant, and was also consistently noted for the other species, namely *K. lactis*, *K. dobzhanskii,* and *K. aestuarii*.

*Pichia*. Thirty-two assemblies belonging to the Pichia genus and included in the analysis were grouped into 13 species. Similar to the *Kazachstania* genus, some species were represented by several assemblies, and others were represented by only one assembly. However, boundaries between *Pichia* species were defined by ANI values below 82%. Genomes of the *P. kudriavzevii* species, the most numerous, displayed pairwise identity values ranging from 98.90% to 100%. The case of *P. membranifaciens* is noteworthy, as only two assemblies belonging to the same species showed an identity value of 85.74%. Again, this observation suggests possible misidentification of isolates or complex genetic interchange events.

*Rhodotorula*. The *Rhodotorula* dataset encompassed 132 genomes, with most of them belonging to *R. mucilaginosa* (101 assemblies) and *R. toruloides* (18 genomes). FastANI analysis defined clear boundaries between species, with ANI values below 80%. For species with multiple assemblies, such as *R. mucilaginosa*, *R. toruloides*, *R. kratochvilovae*, and *R. paludigena*, a same-species cutoff of >95% was observed. However, exceptions were noted, particularly in 7 out of 101 strains within *R. mucilaginosa*, displaying ANI values near 90%. This value suggests the potential existence of hybrids between *R. mucilaginosa* and other species, or the misidentification of a different as-yet-uncharacterized species, especially considering that the identity values for the other 94 strains were above 95%.

*R. toruloides* presents a complex scenario, with higher variability. Indeed, NBRC10032 (GCA_007990605.1) exhibited very low ANI values (below 85%) compared with all the other strains. The remaining strains appeared to be divided into two clusters, with identity values over 96% in one cluster and around 90% in the other. The first cluster, which included the reference strain NP11 (GCA_000320785.2), was made up of GCA_000258745.1, GCA_000320785.2, GCA_000988805.1, GCA_001542305.1, GCA_001600115.1, GCA_001600155.1, GCA_003234015.1, and GCA_016808315.1 The second cluster consisted of GCA_000222205.2, GCA_000988875.1, GCA_000988875.2, GCA_001255795.1, GCA_001456015.1, GCA_001542265.1, and GCA_001600135.1. This divergence suggests potential misidentification based on unique genetic markers. Interestingly, GCA_000222205.2, belonging to the second cluster, is documented as a *R. glutinis* genome, despite its assignment to the *R. toruloides* taxon in the NCBI Genome database. However, it exhibited very low identity with assemblies assigned to the *R. glutinis* species in the NCBI Genome database (Appendix A), underscoring inconsistencies in species delineation, which impact database information. Furthermore, four *R. toruloides* assemblies, GCA_001600215.1, GCA_005387725.1, GCA_007990605.1, and GCA_016808315.1, showed ANI values below 90% with all the others belonging to the same species, hinting at the possibility of belonging to a distinct yet unaccounted-for species, a hypothesis that warrants experimental confirmation.

*Saccharomyces*. This largest dataset consisted of the assemblies from the *Saccharomyces* genus encompassing seven species, with *S. cerevisiae* being the most prevalent. FastANI successfully identified species boundaries, especially for *S. cerevisiae*, *S. eubayanus*, *S. mikatae*, and *S. paradoxus* (Figure 3C, Appendix A). For these species, a cutoff value of 95% identified genomes belonging to the same species. The comparison between *S. eubayanus* and *S. uvarum* genomes showed ANI values of approximately 92%, indicating a close relationship between the two species, as corroborated by previous phylogenomic analyses [35]. Nevertheless, the cutoff of 95% remains effective for identifying genomes of *S. eubayanus* or *S. uvarum*. 

Greater variability was observed among the 23 *S. kudriavzevii* genomes. Indeed, FastANI analysis revealed pairwise identities between genomes of the same species ranging from 92% to 100%. When comparing ANI values between the assembly of the reference strain *S. kudriavzevii* CR85 (GCA_003327635.1) and the other 22 strains, only six of them displayed identity values below 95%. Further studies should be performed to ascertain whether the typical ANI value for intraspecies *S. kudriavzevii* can be established at 92%, or if ANI values are influenced by hybridization events, horizontal gene transfer, or purity of the sequenced DNA sample. Nonetheless, FastANI effectively discriminates this species from the others, as the identity values between them were below 81%.

As observed for other genera, certain ambiguous scenarios were identified. For example, *S. eubayanus* GCA_013181345.1 exhibited ANI values of around 90–92% with both *S. cerevisiae* and *S. uvarum*, suggesting the possibility of its being a hybrid. In contrast, the discriminating value was 98% among other *S. eubayanus* genomes. 

*Schizosaccharomyces*. Despite the limited number of genomes analyzed, with *S. pombe* being the predominant species, FastANI demonstrated a reliable ability to distinguish between different species. Specifically, among the *S. pombe* species, ANI values were notably high, around 99%. The pairwise identity values between *S. pombe* assemblies and assemblies belonging to the other species were considerably low, below 80%. FastANI identified an inconsistency between two strains belonging to the *S. japonicus* species, exhibiting a reciprocal identity below 90%.

*Starmerella*. Despite the low number of *Starmerella* genomes (20 genomes and 12 species), the analysis confirmed that the same species ANI value in this dataset was greater than 97%. Clear boundaries between species were evident, as the ANI between different species remained under 85%. Only the strain *S. bombicola* PYCC 5882 (GCA_003033785.1) had a lower ANI value compared with other *S. bombicola* genomes. This discrepancy may be attributed to experimental artifacts stemming from the lack of purity in the original sample, influencing accurate species delineation, or to the above-mentioned genomic element interchanges.

*Torulaspora*. Despite the limited number of available assemblies, this genus showed a complex structure. Clear boundaries between species were evident, as the ANI values between different species consistently remained below 90%. For some species, such as *T. microellipsoides*, *T. franciscae*, and *T. delbrueckii*, the same-species identity value exceeded 95%. However, inconsistencies were observed among the *T. pretoriensis* species. 

Notably, five assemblies belonging to *T. pretoriensis* displayed lower identity values with other assemblies from the same species. Given that these genomes were part of the same project (Bioproject PRJNA623867) and that no public peer-reviewed articles were associated with it, a misidentification can be speculated. The relatively low sequencing coverage (approximately 40×) is unlikely to impact the results, as other assemblies with consistent ANI values had the same coverage.

*Yarrowia*. The discriminating ANI value for *Yarrowia* species was extremely high, with ANI values between genomes belonging to different species consistently below 90%. Conversely, within the same species, especially in the case of *Y. lipolytica*, represented by 24 genomes, the identity values were very high, exceeding 99%. However, we observed that some strains were assigned to the wrong species. Indeed, the NCAIM 3590 strain (GCA_003571375.1) appeared as an outlier, displaying high ANI values (99.5%) with *Y. bubula* strains, suggesting a potential misidentification. Additionally, assembly GCA_001069125.1 represented a unique case: it was identified as *Y. lipolytica* but presented high FastANI identity values only with some of the other *Y. lipolytica* assemblies and one of the *Y. bubula*; this suggested a potential hybridization event. However, based on D1/D2 results, it is classified as *Y. lipolytica*, which may explain the classification of this strain in the database.

### 3.2. Species Delineation Based on In Silico Extraction of D1/D2 Region and MAOG

The other objective was to evaluate the effectiveness of genome-based-D1/D2 region analysis and multigenic alignment comparison (MAOG) in the determination of species and to compare the results obtained from these analyses with those obtained by FastANI. 

Despite the analysis being applied to the entire dataset of 644 genomes, the D1/D2 analysis has been performed only on 404 assemblies (62.7%), because the D1/D2 region was not found in the remaining 240 assemblies. Similarly, the MAOG method was applied to 484 assemblies out of 644, constituting 75.2% of the total, of which at least 200 orthologous genes were identified from the BUSCO fungi database (see Methods section for more details). Overall, 290 genomes were investigated using both the D1/D2 and MAOG methods, with their distribution among genera described in Table 2. Identity matrices with pairwise identities were generated using the results obtained from the D1/D2 region alignments and MAOG, respectively (see Appendix A for complete data).

When analyzed with the in silico extracted D1/D2 region, the similarity ranges within the same species were found to range from 97% to 100%, with a prevalence of 100% identity values (Figure 4). 

Additionally, we noted cases where the identities between two different species were exceptionally high, exceeding 99% (Appendix A). For example, within *Hanseniaspora* genus, *H. clermontiae* and *H. uvarum* showed an identity value of 99.51%, and an identity value of 99.67% was observed between *H. guilliermondii* and *H. opuntiae* (Figure 5A). Similar values were also observed in *Debaryomyces* and *Saccharomyces* genera. Specifically, *D. prosopidis* demonstrated a similarity of 100% with *D. fabryi* and 99.84% with *D. hansenii* (Figure 5B). Moreover, according to published evidence, *S. paradoxus* and *S. cerevisiae* are close species, showing identity values of more than 99% [42,43] (Figure 5C). 

Concerning MAOG, overall, same-species identity values were observed to vary, ranging from 90% to 100%, depending on the genera. The inter-species boundaries were clear and defined by identity values lower than 90% (Figure 6). 

### 3.3. Comparative Analysis between D1/D2 Sequence Alignments, MAOG, and ANI

We evaluated FastANI accuracy in assigning a yeast strain to a known species within each selected genus and compared it to the other two methods. The evaluation was conducted on a subset of the dataset, specifically on 290 genomes common to all three methods out of the total 644 selected assemblies (Table 1).

By analyzing the identity values derived from the three methods, comparable distributions were observed. Figure 7 shows how values are grouped into several peaks, illustrating a stratified distribution of identity values at different levels rather than two populations corresponding to conspecific and non-conspecific strains. The far-right peak corresponds to identity values between assemblies from the same species, while the other peaks correspond to different levels of distance among species. For instance, in the case of the *Saccharomyces* genus, the second peak contains the values of very close species such as *S. cerevisiae* and *S. paradoxus* or *S. uvarum* and *S. eubayanus* and the third one shows the relationship between more distant species (Appendix A). Although the clustering of these relationships around four main values may have been influenced by the distances among species in our dataset, it is worth noting that these peaks could also potentially include identity values from hybrid genomes, which, although less likely, were still relevant.

Notably, FastANI outperformed the other two methods in terms of discrimination (Figure 4). Indeed, in the other two methods, particularly the D1/D2 method, the first and second peaks showed less separation and a larger area of overlap (also see Appendix A). Of note, while discrimination was clearly evident for MAOG and FastANI methods, the identity range for species assignment using the D1/D2 region was so small that establishing a clear conspecificity threshold was challenging for some species.

Paired *t*-tests, employing Welch modification, were conducted to compare the identity values obtained by FastANI on assemblies from the same species with those obtained by each of the other two methods. The results indicate a statistically significant difference (*p*-values < 2.2 × 10^−16^), with Cohen’s D values of 1.26 and 1.18 for FastANI compared to the D1/D2 region and MAOG, respectively. This, together with the peak distributions above, reinforces the conclusion that FastANI is more capable of delineating species.

A cutoff was then established for each method to differentiate assemblies belonging to the same species from those that do not. This was achieved by calculating the local minimum separating the two main peaks of the density curves. The cutoff identity value on the *X*-axis was higher for FastANI than for MAOG (0.93 and 0.92, respectively). Importantly, the Y coordinate was notably lower in FastANI than in MAOG (0.21 and 1.14, respectively), indicating a lower probability of finding ANI values in regions in which the delineation within the assessed species is unclear (Figure 4 and Figure 7).

Principal component analysis (PCA), incorporating identity values obtained with each of the methods as variables describing each pair of genomes, demonstrated that FastANI identity values provided superior explanatory power compared to the other two methods in distinguishing between pairs of genomes within the same species and those from different species (Appendix A), closely followed by MAOG. Moreover, to assess statistically significant differences among identity values obtained by the different methods, a Wilcoxon test was performed for each genus (Appendix A). Both methods confirmed the considerable distance between themselves and the D1/D2-based one. Moreover, MAOG and FastANI, while similar for most genera, showed distinct behavior for some of them (Appendix A, Appendix A). Notably, data for *Hanseniaspora* highlighted the most statistically significant difference between FastANI and MAOG, showing FastANI’s superior power for species delineation. This distinction, albeit to a lesser extent, was also observed for *Rhodotorula*, *Saccharomyces*, and *Torulaspora*. For the remaining genera, the ability of the two methods to delineate species was comparable.

### 3.4. Exploring Species Delineation in Saccharomyces Hybrid Genomes 

Based on the results above, we noticed that certain assemblies within some genera exhibited ANI values within the boundaries of conspecificity. This finding led us to hypothesize that these cases involved strains that had undergone genomic rearrangements such as hybridization. To investigate this event, we applied the three different methods to another dataset including genomes of *Saccharomyces* natural hybrids (Appendix A). 

The alignment of the D1/D2 fragment was performed on 64 out of 77 hybrid assemblies. In the case of MAOG, the analysis was carried out on 73 of the hybrid assemblies that had passed the quality filter of identifying more than 200 orthologs from the Augustus database, as explained in the Material and Methods section. Then, we calculated the identity values between them and each of the assemblies belonging to four *Saccharomyces* species, in particular, *S. cerevisiae*, *S. eubayanus*, *S. kudriavzevii*, and *S. uvarum*. The results illustrated in Figure 8 show that the analysis with the D1/D2 region was not able to distinguish between hybrids and pure species. Indeed, identity values between pure species and hybrids were always above the threshold. On the contrary, identity values calculated between pure species and hybrids by MAOG were below the threshold as long as the percentage of the parent’s genome dominated considerably over the others. In this case, the assignment of a hybrid strain to a pure species was obtained for hybrids that contained more than 75% of parent species for *S. uvarum* and *S. eubayanus* and more than 85% for *S. cerevisiae*. On the other hand, the FastANI method tended to produce identity values surpassing the conspecificity threshold of 95% when the contribution of a parent species to the hybrid genome reached 25% or more. 

## 4. Discussion

To date, yeast taxonomy is still a challenging and evolving field [44]. Currently, the workflow for taxonomic delineation consists of an integrative approach, including culture-based methods, molecular techniques for DNA barcoding and phylogenetics, and, in some cases, whole genome analysis [14,15,16,45,46]. Given the absence of a definitive method for yeast species delineation [46] and considering the ongoing advances in genome sequencing technologies and bioinformatics methods, this study suggests that the rapid free-alignment genome-based comparison tool FastANI [29] should be integrated into a comprehensive strategy to classify strains into a known species. The selected dataset included 12 yeast genera, for each of which the assemblies of all species available in the NCBI Genome database were considered. As already commented by others [45], it is important to note that NCBI, as it is not a repository specifically curated for the quality of deposited genomes nor for assigning a strain to a species, may contain genomes with different degrees of completeness or classified with incorrect taxonomies. These genomes may have been obtained through different types of sequencing (short reads, long reads, or both). As is well known, short- and long-read sequencing have different characteristics in terms of accuracy and ability to cover the genome. While short-read technology is more accurate it requires high coverage to obtain long DNA sequences, whereas long-read technology is able to cover large areas of the genome, without guaranteeing accuracy comparable to the short-read technology. The choice of sequencing technology is not the only aspect that impacts the quality of the assembly. Other parameters, including the type of platform (e.g., PacBio or ONT), the read length, the coverage depth, and the type of assembly tool, must be considered, and finding the right combination is not easy [47]. Most of the genomes in our dataset have been sequenced using Illumina technology, which leads to the question of whether the assemblies to be analyzed are sufficiently complete to produce reliable results in assigning species. The *in silico* analysis of D1/D2 domains of the LSU, a region traditionally accepted to identify the species in yeasts [43,48,49], has probably been affected by the quality of genomes. Indeed, we observed the impossibility of extracting the selected region from all of the assemblies analyzed in the study. This issue could be attributed to technical reasons linked to the choice of NGS technology or library preparation, or due to an inadequate coverage of raw data. The analysis of the LSU sub-region of a restricted number of genomes highlighted the limited ability of this method to distinguish between strains belonging to distinct species. It is crucial to acknowledge that the extracted D1/D2 sequences in the present work probably contained methodologically derived artifacts, even if this limitation is inherent to the original method as well. Indeed, although Kurtzman and colleagues proposed that conspecific strains showed less than 1% nucleotide divergence in this region [48,49], some different species from genera such as *Debaryomyces* or *Hanseniaspora*, for example, showed identity values over 99.50% [29]. In *Hanseniaspora*, genome-based studies have shown that this genus is composed of two lineages, a fast-evolving lineage (FEL) and a slow-evolving lineage (SEL), that have differing evolution rates. In FEL, the comparison of the D1/D2 sequence was demonstrated to be ineffective for species discrimination between *H. opuntiae* and *H. guilliermondii* and between *H. meyeri* and *H. clermontiae*, in line with our results [17]. The challenge of distinguishing between species using D1/D2 is widely acknowledged, underscoring the notion that D1/D2 or other markers with similar characteristics should not be relied upon alone as the primary method for species differentiation. Instead, these markers should be employed to help refine the identification of genera to which a newly isolated yeast strain might belong [50,51].

MAOG was the other reference method used in this study. Again, we observed the impossibility of analyzing the full dataset of assemblies because some did not present enough completeness to reach the minimum number of orthologous needed for the analysis. This may have been due to the excessive fragmentation of contigs caused by the short-read technologies, which resulted in an incomplete annotation, an issue already identified by other authors [52]. However, unlike the results obtained with the D1/D2 fragment, the results obtained with the MAOG method were comparable to those with FastANI (Figure 7), despite being less precise and more time-consuming.

Here, ANI computation with the alignment-free tool FastANI proved to be an easy and quick way to analyze genomes with different degrees of completeness and quality. ANI has been employed relatively little so far to establish the criteria to be used in species delineation [3,27,53,54], but its application is becoming more widespread as an aid in species delineation and identification [55,56,57,58]. Libkind et al. (2020) conducted a review of applying genome analysis, including the ANI calculation, to yeast taxonomy. The study provided some examples in which identity values in D1/D2 and ITS regions were compared with those calculated by ANI, considering one representative genome among genera of Basidiomycetes and Ascomycetes [3]. From the analysis of the reviewed cases, the authors show an overall intra-species cutoff of 95% and also suggest that this analysis may be useful within a holistic species determination approach. Čadež used an ANI method to study the relationships among *Hanseniaspora* species and in particular, to identify and describe *Hanseniaspora smithiae* as a new species, which showed ANI values of 87% in comparison with the close species *H. valbyensis* [53]. Similarly, the description of the new species *Hanseniaspora menglaensis* has been supported by the use of ANI analysis in comparison to other species within the same genera, highlighting a clear demarcation as a separate species [59]. Wibberg and colleagues used ANI to support the study of the taxonomic classification of some species in the Hypoxylaceae family [54]. Only Lachance and colleagues have furthered the analysis by using the available *Metschnikowia* genomes and establishing the 95% ANI value as a reliable guideline for species delineation [27]. 

The present study follows the framework already outlined by the cited studies, showing how FastANI can support the delineation of the genetic distance of species in yeasts. To our knowledge, this is the first comprehensive study on the use of ANI as a method to assign a strain to a species in different yeast genera. By evaluating overall ANI values, a common threshold of 94–95% emerged across all the genera analyzed in this study, in agreement with the findings of others [6,47]. Regarding intra-genus data, we can observe that for some genera, such as *Candida* and *Hanseniaspora*, higher cutoffs (around 96%) may be appropriate. It is important to acknowledge exceptions to these thresholds, as noted for some of the genera studied in this work. It should be noted that we cannot exclude the impact of sequencing technology on the definition of the ANI cutoffs proposed in this study. As mentioned previously, the pros and cons concerning the choice of sequencing platform and pipeline to be applied for obtaining the assembly and its analysis must be considered. Although the work on the application of FastANI to determine cutoffs between bacterial species Jain et al. [29] showed that this tool is not affected by genomic completeness, it would be important to investigate the influence of the sequencing platform and the tools used for assembly on the results of ANI analysis.

In addition, FastANI may be used to deduce complex genome modification events such as hybridization and to identify experimental artifacts such as species misidentification. In the present work, very high identity values between assemblies that were supposed to belong to different species were observed, suggesting the presence of hybrid organisms. Such intermediate values were lower than those obtained from the same species comparison and higher than those resulting from the comparison with different species. This is quite common in the *Saccharomyces* genus, from which a variety of hybrid strains have been isolated, especially in industrial environments, including *S. pastorianus* in beer [60] and *S. cerevisiae* × *S. uvarum* and *S. cerevisiae* × *S. kudriavzevii* in wine fermentations [61,62]. When we tested the three methods on a dataset composed of previously characterized *Saccharomyces* hybrids, the analyses indicated that a considerably larger portion of a species genome would be required to achieve noticeably higher identity values with both the MAOG and the FastANI methods (Figure 8). The MAOG method appears to be more resilient to very high identity values between hybrids and pure species, which would thus avoid erroneous delineation of these assemblies. Nevertheless, some instances, especially in the case of *S. uvarum*, exhibited identity close to the conspecificity threshold defined above. The reason why the rest of the species in the study behaved differently is difficult to deduce from our dataset. Interestingly, the FastANI method tended to produce identity values surpassing the threshold in hybrids when the contribution of a parent species to the genome was 25% or more (Figure 8). This feature may be strategically employed in a pipeline for species delineation, particularly when FastANI identity values surpass the threshold for more than one species, markedly increasing the probability of classifying the assembly as a hybrid. 

## 5. Conclusions

As genomic data continue to proliferate, tools like FastANI, not used as a standalone but as part of a multifaceted taxonomic strategy, will contribute to refining our understanding of yeast diversity and taxonomy, thus paving the way for a more accurate and nuanced species classification. In this work, we have analyzed an extensive dataset and proposed cutoffs that can be a guide for assigning a yeast strain to a known species. Moreover, we suggest using FastANI on data sets that are as inclusive as possible, to explore comparisons with different species within the same genus. This analysis should be part of a holistic approach that includes phenotypic and morphological analyses.

Our findings also emphasize the need for such adaptable approaches in the face of complexities and potential hybridization scenarios.

## Figures and Tables

**Figure 1 jof-10-00646-f001:**
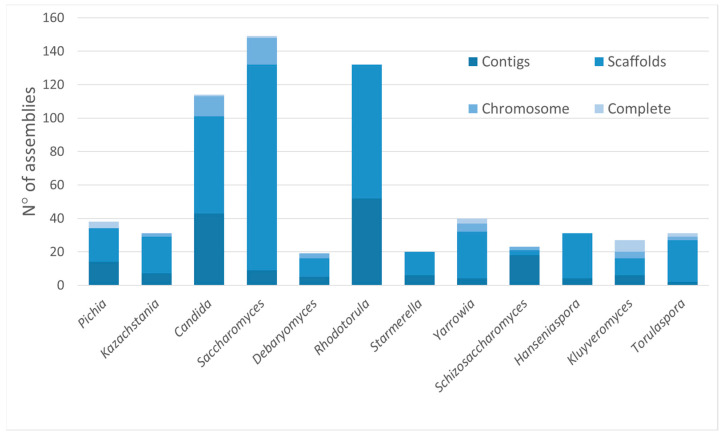
The completeness status of assemblies belonging to the analyzed dataset. Contigs are sequence fragments; scaffolds are contigs connected by spanned gaps; chromosome status indicates that one or more chromosomes could be complete, and other sequences could be made by scaffold or contigs; and complete status refers to the entire chromosome sequences without gaps, including plasmid(s), if present.

**Figure 2 jof-10-00646-f002:**
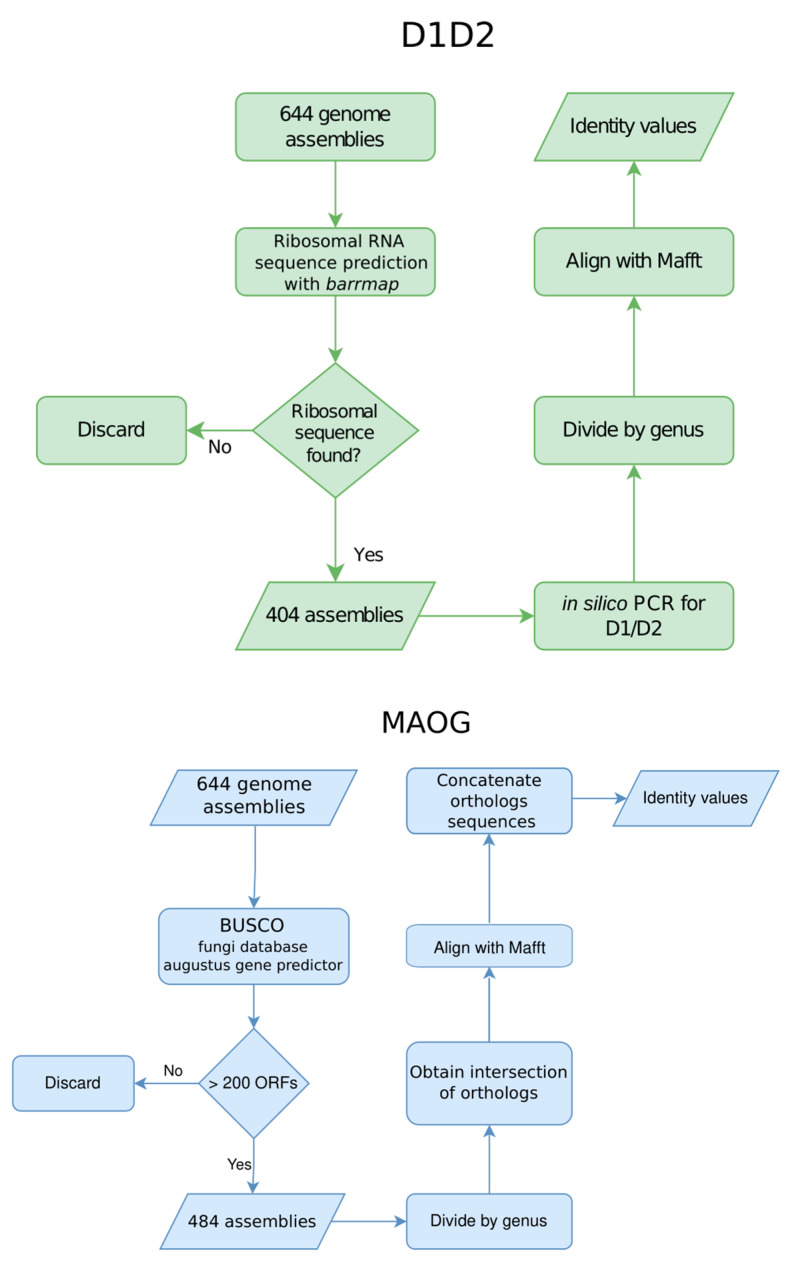
The workflow and tools used to analyze the dataset of assemblies using the three methods, D1/D2 region alignment, multiple alignment of orthologous genes (MAOG), and FastANI.

**Figure 3 jof-10-00646-f003:**
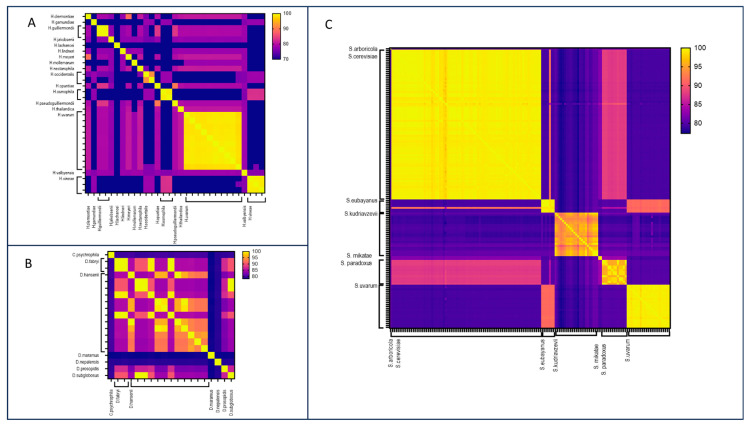
Heatmaps of the FastANI ID values obtained from the pairwise comparison of the *Hanseniaspora* (**A**), *Debaryomyces* (**B**), and *Saccharomyces* (**C**) assemblies.

**Figure 4 jof-10-00646-f004:**
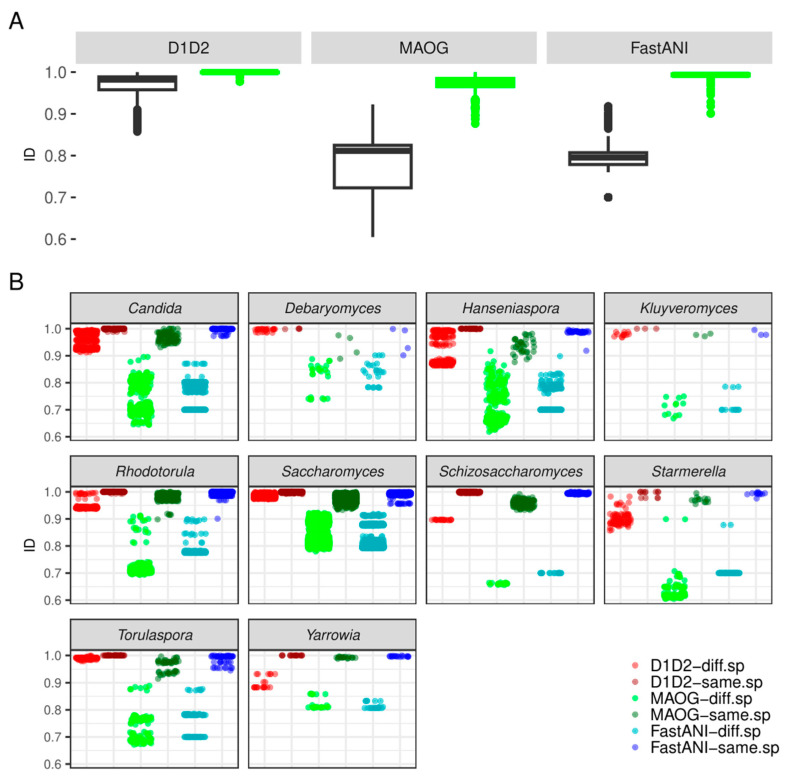
Comparison of discriminating power between D1/D2, MAOG, and FastANI methods, considering the dataset composed of assemblies analyzed by all three methods (290 genomes). (**A**) Boxplots of ID (identity) values from pairwise combinations of assemblies belonging to the same (green) or different (black) species for each method. (**B**) The scatterplot describes the genus-by-genus distribution of identity values from pairwise combinations of assemblies belonging to the same or different species obtained with the three methods (D1/D2 region, MAOG, and FastANI). Points are colored by method and conspecificity. In the legend key, “diff.sp“ means assemblies belonging to different species according to the NCBI Genome database and “same.sp” means the assemblies belonging to the same species according to the NCBI Genome database.

**Figure 5 jof-10-00646-f005:**
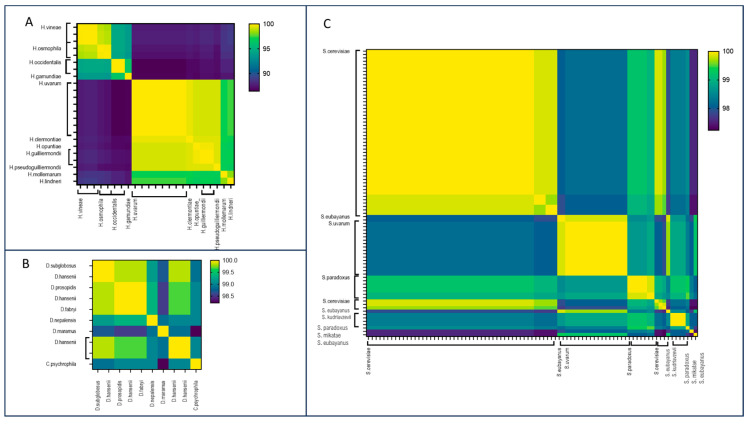
Heatmaps of the identity values obtained from the pairwise comparison between D1/D2 sequences of the *Hanseniaspora* (**A**), *Debaryomyces* (**B**), and *Saccharomyces* (**C**) assemblies were analyzed.

**Figure 6 jof-10-00646-f006:**
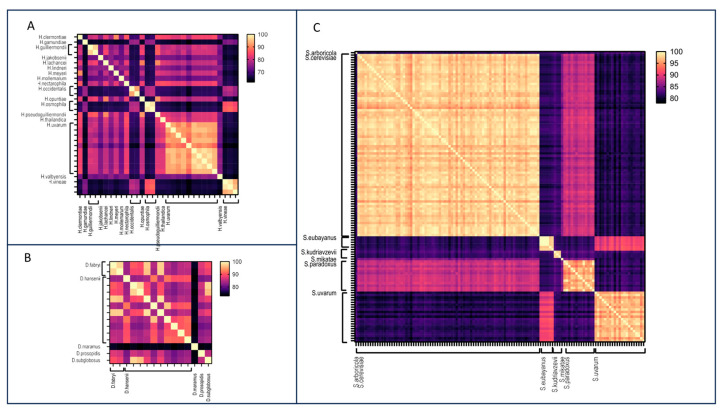
Heatmaps of the identity values obtained from the pairwise comparison obtained with the MAOG method for *Hanseniaspora* (**A**), *Debaryomyces* (**B**), and *Saccharomyces* (**C**) assemblies.

**Figure 7 jof-10-00646-f007:**
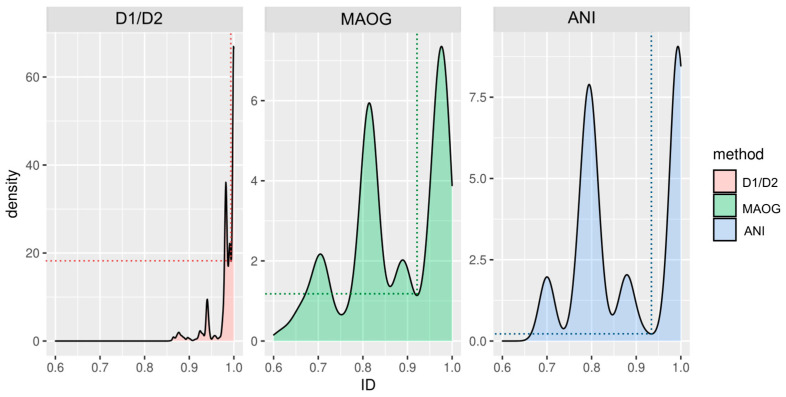
Density estimation of the pairwise identity values among 290 assemblies was obtained using the D1/D2, MAOG, and FastANI methods. Only identity values between assemblies belonging to the same genus were considered. Curves are a smoothed version of the histogram obtained by the geom_density function of the ggplot2 package in R. The use of these data distributions allowed for the calculation of the local minima between the two peaks on the far right, identified by dotted lines.

**Figure 8 jof-10-00646-f008:**
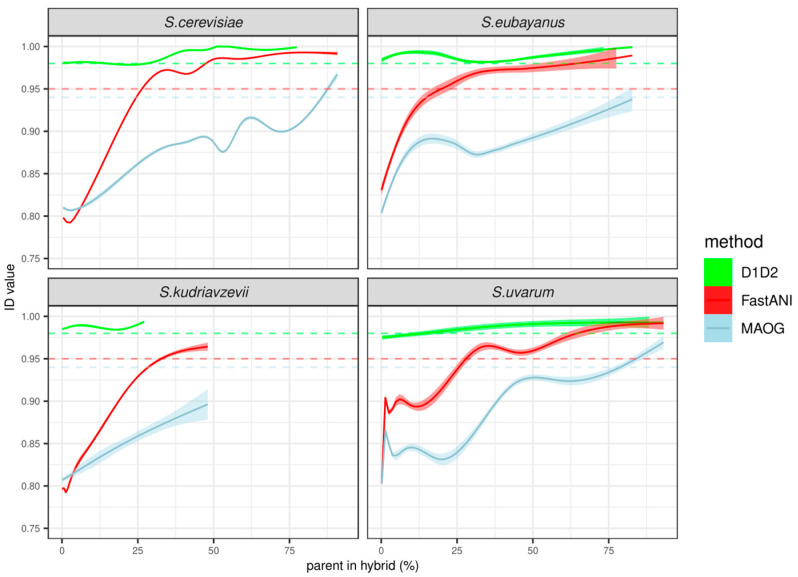
Identity values between assemblies from interspecific *Saccharomyces* hybrids and assemblies from pure species. Each panel depicts the identity values versus the percentage of the corresponding species shown in the titles. Trend lines (continuous) show the data smoothing performed by the geom_smooth function of the package ggplot2 in R, using a generalized additive model (gam), except for *S. eubayanus*, in which the LOESS method was used due to a lack of data. Shadows show confidence intervals of the estimated formula (α = 0.95). Dashed lines represent conspecificity thresholds for each of the three methods: 0.995, 0.926, and 0.937 for D1/D2, MAOG, and FastANI, respectively.

**Table 1 jof-10-00646-t001:** The identity values cutoffs were calculated by FastANI (Average Nucleotide Identity calculation) among the species belonging to the twelve considered genera. The within-species cutoff is the ANI value that identifies strains within the same species; conversely, the different-species cutoff is the ANI value specifying assemblies belonging to different species.

Genus	Within-Species Cutoff	Between-Species Cutoff
*Candida*	>96%	<88%
*Debaryomyces*	>92%	<90%
*Hanseniaspora*	>97%	<90%
*Kazachstania*	>98%	<89%
*Kluyveromyces*	>95%	<79%
*Pichia*	>98%	<82%
*Rhodotorula*	>95%	<80%
*Saccharomyces*	>92%	<81%
*Schizosaccharomyces*	>98%	<80%
*Starmerella*	>97%	<85%
*Torulaspora*	>95%	<90%
*Yarrowia*	>98%	<90%

**Table 2 jof-10-00646-t002:** Number of assemblies belonging to the dataset, divided by genus (see Appendix A for all accession numbers), and number of assemblies analyzed by each method and by all three methods.

Genus	Total N° of Assemblies	N° of Assemblies Analyzed by FastANI	N° of Assemblies Analyzed by MAOG	N° of Assemblies Analyzed by D1/D2 Alignment	N° of Assemblies Analyzed by All Three Methods
*Pichia*	32	32	32	23	23
*Kazachstania*	28	28	20	17	16
*Candida*	114	114	104	34	29
*Saccharomyces*	148	148	121	86	80
*Debaryomyces*	19	19	15	10	8
*Rhodotorula*	132	132	47	115	36
*Starmerella*	20	20	20	14	14
*Yarrowia*	39	39	28	15	9
*Schizosaccharomyces*	23	23	23	20	20
*Hanseniaspora*	31	31	31	23	23
*Kluyveromyces*	27	27	20	20	14
*Torulaspora*	31	31	23	27	18
Total	644	644	484	404	290

## Data Availability

The original contributions presented in the study are included in the article/Appendix A, further inquiries can be directed to the corresponding author.

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
