# Peer review of "Evaluating the Genome-Based Average Nucleotide Identity Calculation for Identification of Twelve Yeast Species"

_jof, 2024, doi:10.3390/jof10090646_

Round 1

Reviewer 1 Report

In this manuscript, the authors propose the use of comparative genomics approaches as an additional tool in yeast taxonomy. Of the three approaches tested - D1/D2 comparison, FastANI and MAOG, FastANI gives the bests results being able to differentiate between species within a genus and also identify genome arrangments by hybridization with a same species. The authors use large sets of genomic assemblies for their analysis so their work will definitely be very useful all the yeast community. 

1/ Introduction

- Line 42: ¿has BEEN linked? 

- Line 44: It is not clear what the authors mean with "the definition of biodiversity of yeasts", please reformulate. 

- Line 53: AND the actin gene 

- Line 78: Define LSU and then use the abbreviation in all the manuscript 

2/ Materials and methods 

- Line 115: ... because of their numerous applications in biotechnology... instead of because of their role in the biotechnology field.

- Paragraph 2.2 Include the reference and version of all softwares that are used as well as the parameters. 

3/ Results 

In the ANI analysis (3.1), the % of coverage must be included in addition to the % of identity. 

Table 1: Add a column with the % identity between species in addition to within species 

Table 2. Add a column to inform the type of sequening method i.e. short fragments or large fragments and discuss the implications of using large or short fragments sequencing on the cutoff percentages defined by the authors. 

4/ Discussion

Please also cite the following two papers: 

a) Freitas, L. F. D. et al. (2020): 

Yeast communities associated with cacti in Brazil and the description of Kluyveromyces starmeri sp. nov. based on phylogenomic analyses. Yeast (Chichester, England), 37(12), 625–637.

b) Lozano-Aguirre, L. et al. (2024)

Draft genomes of four Kluyveromyces marxianus isolates  retrieved from the elaboration process of henequen 

(Agave fourcroydes) mezcal. Microbiology resource announcements, 13(3), e0086123.

Lines 534-541: the authors should discuss a little more the results obtained by other authors using ANI and how their results compare with these other studies. 

5/ Conclusions 

 The conclusion is very general, the authors could give more precise conclusions on ANI, and emit recommendations for practical cut off values to be used. 

x

Author Response

REVIEWER 1

Major comments

In this manuscript, the authors propose the use of comparative genomics approaches as an additional tool in yeast taxonomy. Of the three approaches tested - D1/D2 comparison, FastANI and MAOG, FastANI gives the bests results being able to differentiate between species within a genus and also identify genome arrangments by hybridization with a same species. The authors use large sets of genomic assemblies for their analysis so their work will definitely be very useful all the yeast community. 

1/ Introduction

- Line 42: ¿has BEEN linked? Thank you. We corrected the verb.

- Line 44: It is not clear what the authors mean with "the definition of biodiversity of yeasts", please reformulate. Thank you. We modified the sentence, in a clearer way.

- Line 53: AND the actin gene Thank you, we added AND.

- Line 78: Define LSU and then use the abbreviation in all the manuscript Thank you. We have defined LSU as “ the variable domains D1 and D2 of the nuclear large subunit ribosomal RNA (LSU rRNA)” in lines 47-48.

2/ Materials and methods 

- Line 115: ... because of their numerous applications in biotechnology... instead of because of their role in the biotechnology field.

Thank you, we corrected the sentence as suggested.

- Paragraph 2.2 Include the reference and version of all softwares that are used as well as the parameters. 

Thank you. We added the versions of tools we used.

3/ Results 

In the ANI analysis (3.1), the % of coverage must be included in addition to the % of identity.

Thank you for the comment. To specify this, we added details about the command (new line 193-194: minimum fraction of shorter genome coverage of 50%) and an additional file (new Table S4) to be included in supplementary files with the one-to-one identity, the of number of fragments matched between the two strains and the number of fragments in total between the two strains.

Table 1: Add a column with the % identity between species in addition to within species 

Thank you for the comment. We added the column of identity values between species.

Table 2. Add a column to inform the type of sequening method i.e. short fragments or large fragments and discuss the implications of using large or short fragments sequencing on the cutoff percentages defined by the authors. 

Thank you for the comment. In Table S1, which contains the list and characteristics of all genomes considered in the dataset, we added a column about the sequencing technology used to obtain raw data. Moreover, we add some discussion about how different sequencing technology may affect the species identification by ANI analysis. Please, see new lines 581-597 and 662-669.

4/ Discussion

Please also cite the following two papers: 

  1. a) Freitas, L. F. D. et al. (2020): 

Yeast communities associated with cacti in Brazil and the description of Kluyveromyces starmeri sp. nov. based on phylogenomic analyses. Yeast (Chichester, England), 37(12), 625–637.

  1. b) Lozano-Aguirre, L. et al. (2024)

Draft genomes of four Kluyveromyces marxianus isolates  retrieved from the elaboration process of henequen 

(Agave fourcroydes) mezcal. Microbiology resource announcements, 13(3), e0086123.

Lines 534-541: the authors should discuss a little more the results obtained by other authors using ANI and how their results compare with these other studies. 

Thank you for the suggestions. We added the suggested references and also other papers in which the species delineation was supported by ANI analysis, and hence we added some discussion about them. Please, see new lines 631 and following.

5/ Conclusions 

 The conclusion is very general, the authors could give more precise conclusions on ANI, and emit recommendations for practical cut off values to be used. 

Thank you. We improve the part of conclusion. Please, see new lines 670-674.

Reviewer 2 Report

This manuscript is a straightforward comparison of two established species classification methods (D1/D2 sequences of LSU rRNA, multiple alignment of orthologous genes/MAOG) with the relatively novel method called FastANI (FastAverage Nucleotide Identity) using hashed sequence data to minimize computational demands. A total of 644 genome assemblies covering 12 yeast species were analyzed (between 19 - 148 depending on the genus) using fastANI (only 404 and 484 of these assemblies by D1/D2 or MAOG, respectively, due to data quality and/or technical limitations of these approaches). The data presented shows that FastANI is superior or equal to the other methods (especially to D1D2 comparisons) in most aspects. A notable exception is the use of MAOG in the identification of hybrid species. The authors conclude that fastANI is an excellent additional tool in the proper identification of taxa.

The manuscript is competently done and well written in good English. In light of the main goal, it probably can't be avoided that some parts are a bit "laundry-list"-ish. That does not detract from the overall quality.

If I have any suggestions for improvement, it would be in the visual presentation. Fonts are VERY small in some cases, for example in the labeling of the heatmaps in Figure 3. Maybe more space could be devoted to these figures to make the labling legible. It's OK when read as PDF, because the resolution is sufficient to get clearly readable text. But on printouts, which some readers still prefer, it is difficult.

One small correction: lines 93/94 on the second page - there is an article missing. I think it should be "....were the first to use an ANI approach..." or "...were the first to use the ANI approach..."

Author Response

REVIEWER 2

Major comments

This manuscript is a straightforward comparison of two established species classification methods (D1/D2 sequences of LSU rRNA, multiple alignment of orthologous genes/MAOG) with the relatively novel method called FastANI (FastAverage Nucleotide Identity) using hashed sequence data to minimize computational demands. A total of 644 genome assemblies covering 12 yeast species were analyzed (between 19 - 148 depending on the genus) using fastANI (only 404 and 484 of these assemblies by D1/D2 or MAOG, respectively, due to data quality and/or technical limitations of these approaches). The data presented shows that FastANI is superior or equal to the other methods (especially to D1D2 comparisons) in most aspects. A notable exception is the use of MAOG in the identification of hybrid species. The authors conclude that fastANI is an excellent additional tool in the proper identification of taxa.

The manuscript is competently done and well written in good English. In light of the main goal, it probably can't be avoided that some parts are a bit "laundry-list"-ish. That does not detract from the overall quality.

Detail comments

If I have any suggestions for improvement, it would be in the visual presentation. Fonts are VERY small in some cases, for example in the labeling of the heatmaps in Figure 3. Maybe more space could be devoted to these figures to make the labling legible. It's OK when read as PDF, because the resolution is sufficient to get clearly readable text. But on printouts, which some readers still prefer, it is difficult.

One small correction: lines 93/94 on the second page - there is an article missing. I think it should be "....were the first to use an ANI approach..." or "...were the first to use the ANI approach..."

Thank you for your comments. It is true that fonts are sometimes very small, especially in the heatmaps. Unfortunately, as the strains analyzed are many and belong to different species, it is difficult to put the names in a larger font without overlapping them or making the image unclear. For better reading, we recommend zooming in on the image directly and apologize for the inconvenience.

We increase fonts in figure 1 and figure 2.

We corrected the sentence, as suggested.